# Distinct Impact of Doxorubicin on Skeletal Muscle and Fat Metabolism in Mice: Without Dexrazoxane Effect

**DOI:** 10.3390/ijms26031177

**Published:** 2025-01-29

**Authors:** Birgit Van Asbroeck, Dustin N. Krüger, Siel Van den Bogaert, Dorien Dombrecht, Matthias Bosman, Emeline M. Van Craenenbroeck, Pieter-Jan Guns, Eric van Breda

**Affiliations:** 1Department of Rehabilitation Sciences & Physiotherapy, Research Group MOVANT, University of Antwerp, Universiteitsplein 1, B-2610 Antwerp, Belgium; birgit.vanasbroeck@uantwerpen.be (B.V.A.); doriendombrecht@hotmail.com (D.D.); eric.vanbreda@uantwerpen.be (E.v.B.); 2Laboratory of Physiopharmacology, GENCOR, University of Antwerp, Universiteitsplein 1, B-2610 Antwerp, Belgium; dustin.kruger@uantwerpen.be (D.N.K.); siel.vandenbogaert@uantwerpen.be (S.V.d.B.); matthias.bosman@uantwerpen.be (M.B.); 3Research Group Cardiovascular Diseases, GENCOR, University of Antwerp, Universiteitsplein 1, B-2610 Antwerp, Belgium; emeline.vancraenenbroeck@uantwerpen.be; 4Department of Cardiology, Antwerp University Hospital (UZA), Drie Eikenstraat 655, B-2650 Edegem, Belgium

**Keywords:** doxorubicin, dexrazoxane, cachexia, adipose tissue, skeletal muscle, metabolism

## Abstract

The chemotherapeutic agent doxorubicin (DOX) leads to the loss of skeletal muscle and adipose tissue mass, contributing to cancer cachexia. Experimental research on the molecular mechanisms of long-term DOX treatment is modest, and its effect on both skeletal muscle and adipose tissue has not been studied in an integrative manner. Dexrazoxane (DEXRA) is used to prevent DOX-induced cancer-therapy-related cardiovascular dysfunction (CTRCD), but its impact on skeletal muscle and adipose tissue remains elusive. Therefore, this study aimed to investigate the long-term effects of DOX on adipose tissue and skeletal muscle metabolism, and evaluate whether DEXRA can mitigate these effects. To this end, 10-week-old male C57BL6/J mice (*n* = 32) were divided into four groups: (1) DOX, (2) DOX-DEXRA combined, (3) DEXRA and (4) control. DOX (4 mg/kg weekly) and DEXRA (40 mg/kg weekly) were administered intraperitoneally over 6 weeks. Indirect calorimetry was used to assess metabolic parameters, followed by a molecular analysis and histological evaluation of skeletal muscle and adipose tissue. DOX treatment led to significant white adipose tissue (WAT) loss (74%) and moderate skeletal muscle loss (Gastrocnemius (GAS): 10%), along with decreased basal activity (53%) and energy expenditure (27%). A trend toward a reduced type IIa fiber cross-sectional area and a fast-to-slow fiber type switch in the Soleus muscle was observed. The WAT of DOX-treated mice displayed reduced Pparg (*p* < 0.0001), Cd36 (*p* < 0.0001) and Glut4 (*p* < 0.05) mRNA expression—markers of fat and glucose metabolism—compared to controls. In contrast, the GAS of DOX-treated mice showed increased Cd36 (*p* < 0.05) and Glut4 (*p* < 0.01), together with elevated Pdk4 (*p* < 0.001) mRNA expression—suggesting reduced carbohydrate oxidation—compared to controls. Additionally, DOX increased Murf1 (*p* < 0.05) and Atrogin1 (*p* < 0.05) mRNA expression—markers of protein degradation—compared to controls. In both the WAT and GAS of DOX-treated mice, Ppard mRNA expression remained unchanged. Overall, DEXRA failed to prevent these DOX-induced changes. Collectively, our results suggest that DOX induced varying degrees of wasting in adipose tissue and skeletal muscle, driven by distinct mechanisms. While DEXRA protected against DOX-induced CTRCD, it did not counteract its adverse effects on skeletal muscle and adipose tissue.

## 1. Introduction

Over the last few decades, cancer survival rates have significantly improved as a result of better screening, early diagnosis and the development of novel drug therapies. Meanwhile, the number of cancer survivors suffering from short- and long-term side effects of anticancer treatment has increased [1,2]. Classical chemotherapy remains a cornerstone in cancer treatment [3,4]. Among chemotherapeutics, the anthracycline doxorubicin (DOX) is widely used because of its low cost and high efficacy against various cancer types, including solid tumors, hematological malignancies and several pediatric cancers. However, DOX has a rather non-selective mode of action and may exert toxic effects on other tissues [5]. More specifically, cancer therapy-related cardiovascular dysfunction (CTRCD) induced by DOX is an important clinical concern and has been investigated extensively [6,7,8]. Currently, dexrazoxane (DEXRA) is the only approved drug to prevent or reduce DOX-induced CTRCD [9].

Aside from CTRCD, DOX significantly contributes to cancer cachexia, a condition marked by the loss of skeletal muscle (SkM) and adipose tissue (AT) mass, which affects up to 80% of patients with cancer and accounts for 20% of cancer-related deaths [4,10].

SkM and AT serve as crucial energy reservoirs and metabolic regulators, making their structural and functional preservation vital for sustaining metabolic health [11,12]. Clinical research shows that patients with cancer tend to be hypometabolic or hypermetabolic, resulting in energy imbalances, which are associated with adverse outcomes [1,13]. However, knowledge of the effects of chemotherapy on energy metabolism remains inconclusive. According to a systematic review by Van Soom et al., the resting energy expenditure (REE), a key component of the total energy expenditure (TEE), in patients with cancer—regardless of cancer type, stage or chemotherapy regimen—follows a U-shaped curve, with a nadir during mid-treatment and a pinnacle at both the start and end of chemotherapy [1]. Nonetheless, it remains unclear whether this reduction is due to decreased tumor activity resulting from the cancer treatment or a consequence of chemotherapy-induced side effects.

Mechanistically, the DOX-induced deterioration of both SkM and AT function and mass is explained by the excessive generation of mitochondrial reactive oxygen species (ROS), culminating into oxidative stress. This state of oxidative stress impedes ATP production and storage, promotes lipolysis and protein degradation, and inhibits protein synthesis, adipogenesis and lipogenesis [14,15]. However, experimental studies have primarily focused on the acute effects of a single, high-concentration bolus (15–20 mg/kg DOX), with only a few studies examining AT changes [14,16]. Remarkably, the combined impact of DOX on both SkM and AT at the molecular level has not been extensively studied. This research gap is noteworthy since it has been proposed that AT loss may precede and potentially drive SkM wasting [17,18,19]. Consequently, the importance of AT-SkM crosstalk in maintaining muscle homeostasis urges the need for more comprehensive research.

Additionally, SkM wasting may occur in a muscle fiber type-dependent manner. Research has demonstrated that type II fibers are more susceptible to a variety of atrophic conditions, including cancer cachexia [3,14,20]. As such, aside from an in-depth characterization of molecular mechanisms, muscle fiber typing is crucial to guide the development of evidence-based therapeutic strategies [3].

From a clinical perspective, SkM and AT loss severely compromises patients’ quality of life, and the tolerability and efficacy of cancer treatment due to an increased incidence of chemotherapy-related toxicity. This often necessitates dose reductions, leading to less effective treatment and ultimately lower survival rates [14,21]. Currently, the therapeutic potential of DEXRA to mitigate the DOX-induced loss of SkM and AT is yet to be determined. To the best of our knowledge, only Zima et al. have examined the effect of DEXRA on skeletal muscle protease activity in rats following a single bolus of DOX [22].

The present study aimed to investigate the catabolic effects of DOX on SkM and AT mass and function using a 6-week treatment protocol in mice, as well as to evaluate the therapeutic effects of DEXRA. This study builds on our previous finding that DEXRA protects against DOX-induced vascular toxicity in these mice [23].

## 2. Results

### 2.1. DOX Reduced Body Weight, Basal Activity and Metabolic Function, Regardless of DEXRA

DOX-treated mice, with or without DEXRA, experienced significant body weight loss (−7.9 ± 1.2% and −5.9 ± 1.4%, respectively) after 6 weeks. In contrast, CON and DEXRA-treated mice gained body weight (9.8 ± 1.1% and 6.3 ± 1.1%, respectively) (Figure 1B). Overall, 38% of the DOX-treated mice and 13% of the DOX-DEXRA-treated mice were identified as cachectic. The majority of the DOX-treated mice (with or without DEXRA) presented with pre-cachexia.

Similar food intake between groups was observed (Figure 1C). Conversely, basal activity decreased in response to DOX treatment, irrespective of DEXRA, compared to the CON and DEXRA group (DOX vs. CON: −53%; DOX-DEXRA vs. DEXRA: −41%) (Figure 1D). Likewise, EE decreased in both DOX groups compared to the CON and DEXRA groups (Figure 1E). However, after statistically adjusting for key determinants of EE—specifically, TIA and eWAT weights as proxies for fat-free mass (FFM) and fat mass (FM), respectively—using ANCOVA, statistical significance between groups was no longer present. The RER was determined to gain insight into substrate oxidation (i.e., carbohydrates, lipids or proteins). However, no significant differences in RER were observed between groups (Figure 1F). Based on the applied formulas, a main DOX effect for carbohydrate oxidation was found, demonstrating that DOX-treated mice tended to have lower levels of carbohydrate oxidation (Figure 1G). In contrast, lipid oxidation remained unaffected (Figure 1H).

### 2.2. DOX Induced SkM and AT Wasting, Partially Reversed by DEXRA

TIA and GAS wet weights normalized over tibia length were lower in the DOX-treated mice compared to the CON and DEXRA-treated mice after 6 weeks. For both muscles, this ratio was also lower in the DOX-DEXRA-treated group compared to the DEXRA but not the CON group (*p* > 0.05) (Figure 2A,C). Although not reaching statistical significance, the same trend was observed for the SOL (Figure 2B). In both the TIA and SOL, a subtle visual trend was noted where DEXRA positively impacted the muscle-wet-weight-to-tibia-length ratio, thereby partially mitigating the detrimental effect of DOX (Figure 2A,B).

Similarly to the FFM, the eWAT wet weight normalized over tibia length was lower in the DOX-treated mice, irrespective of DEXRA, compared to the CON and DEXRA-treated mice (Figure 2D).

### 2.3. DEXRA Encompassed Complex Effects on Muscle Fiber Type and Muscle CSA

Representative images of the muscle fiber type staining of the TIA and SOL are shown in Figure 3A. The difference in muscle fiber type composition between the two muscles is significant and well accepted. The SOL comprised approximately 30% type I, 50% type IIa and 20% type IIb/x fibers. In contrast, the TIA predominantly consisted of type IIb/x (94%) fibers, with only 5% type IIa and 1% type I fibers. Changes following DOX were found in the SOL but not in the TIA for the distribution of type I and type IIb/x fibers (Figure 3B,C). DOX-treated mice exhibited a higher number of type I fibers than DOX-DEXRA-treated mice. This observation was also evident in comparison to the CON and DEXRA-treated mice, but statistical significance was not reached (*p* > 0.05) (Figure 3C). This finding potentially indicates the fiber type switching to more type I (slow-twitch) fibers as a result of DOX, which could be prevented or reversed by the administration of DEXRA. Finally, an interaction effect was observed for the number of type IIb/x fibers. However, post hoc multiple comparisons revealed no differences between the groups (Figure 3C). Consequently, DEXRA administration in DOX-treated mice may have counteracted the DOX-induced shift toward more type I fibers by promoting a switch to more type IIb/x fibers.

Differences were found in the muscle fiber type-specific CSA of the TIA and the SOL (Figure 3D,E).

Regarding the CSA of type I fibers in the SOL, mice from the DEXRA group demonstrated a higher mean CSA for type I muscle fibers than the DOX group. This effect was also evident compared to the CON group but without reaching statistical significance (*p* > 0.05) (Figure 3E). For the TIA, an interaction effect was found regarding the CSA of the type I fibers. However, post hoc multiple comparisons revealed no differences between the groups (Figure 3D). DEXRA combined with DOX tended to increase the CSA of type I fibers.

For the CSA of the type IIa fibers in the SOL, a main DOX and DEXRA effect was found, with post hoc multiple comparisons revealing differences between the DOX and DEXRA groups (Figure 3E). A similar result for the TIA with regard to the type IIa CSA was observed, demonstrating a main DOX and DEXRA effect, with post hoc multiple comparisons revealing differences between the DOX and DEXRA groups (Figure 3D). In general, for both muscles, the mean CSA of the type IIa fibers tended to slightly decrease in response to DOX and increase in response to DEXRA alone.

Finally, DOX-DEXRA-treated mice demonstrated a lower mean CSA for type IIb/x muscle fibers in the SOL compared to the DOX-treated mice (Figure 3E). This effect was not present in the TIA muscle.

### 2.4. DOX Induced Irreversible Changes in Metabolic Markers and Protein Degradation in SkM, Despite DEXRA

To examine the effects of DOX and DEXRA on glucose and lipid metabolism, RT-qPCR was used to analyze *Pparg*, *Ppard* and *Pdk4* expression in the GAS, along with *Cd36* and *Glut4*, which facilitate free fatty acid (FFA) and glucose uptake, respectively. *Pparg* expression was undetectable, which was not unexpected given its predominant expression in WAT and much lower levels in SkM. Regarding *Ppard* expression, no differences between the groups were noted (Figure 4A). Conversely, *Cd36* expression increased in the DOX group compared to the CON group and, similarly, in the DOX-DEXRA group compared to the DEXRA group (Figure 4B). Likewise, *Glut4* expression increased following DOX treatment in general, without a therapeutic effect of DEXRA. Here, a difference between the DOX and DEXRA groups was also found (Figure 4C). *Pdk4* expression increased following DOX treatment, again with no protective effect from DEXRA. Post hoc multiple comparisons demonstrated differences between the CON and DOX groups, the DEXRA and DOX groups, and the DEXRA and DOX-DEXRA groups (Figure 4D). Finally, two key mediators of the ubiquitin–proteasome system (UPS), *Murf1* and *Atrogin1,* involved in protein degradation, were examined. *Murf1* and *Atrogin1* expression was increased with DOX treatment, irrespective of DEXRA, compared to the CON and DEXRA groups. Post hoc multiple comparisons for both genes revealed differences between the CON and DOX groups and the DEXRA and DOX-DEXRA groups (Figure 4E,F).

### 2.5. DOX Induced Irreversible Changes in Metabolic Markers Suggestive of Impaired Lipid Storage in AT, Despite DEXRA

The same metabolic markers analyzed in the GAS were analyzed in eWAT using RT-qPCR. As such, *Pparg* expression decreased in both DOX conditions in comparison to the CON and DEXRA groups (Figure 5A). Similarly to in the GAS, no differences between the groups were found for *Ppard* expression (Figure 5B). Unlike in the GAS, *Cd36* expression decreased under both DOX conditions compared to the CON and DEXRA groups (Figure 5C). Similarly, a reduction in *Glut4* expression was observed following DOX treatment compared to the CON and DEXRA groups (Figure 5D). Finally, *Pdk4* expression did not differ between the groups (Figure 5E).

## 3. Discussion

DOX is well documented for inducing CTRCD in patients. Additionally, experimental studies, typically involving a single high bolus (15–20 mg/kg of DOX), reported SkM wasting and, in a limited number of studies, AT loss, together contributing to cachexia. In the present study, we examined the impact of long-term (six-week) DOX treatment on both SkM and AT, and the therapeutic potential of DEXRA. Our findings reveal varying degrees of wasting in AT and SkM post-DOX treatment, along with distinct mechanisms regulating metabolic processes and protein balance between the tissues. Remarkably, while DEXRA protected against DOX-induced CTRVD, it had no effect on SkM or AT [23].

DOX induced (pre-)cachexia, as demonstrated by the significant reduction in total body weight, primarily due to the loss of AT and, to a lesser extent, SkM mass [14,24]. This was evidenced by the decreased weight-to-tibia-length ratios for WAT and SkM (independent of muscle type) compared to controls. These results align with other studies highlighted in several literature reviews, albeit somewhat limited, focused on either SkM or AT [3,14,15,16]. Several studies using murine models of cancer cachexia have suggested that fat loss and altered fat metabolism predispose individuals to muscle loss, indicating that fat loss can trigger SkM loss [17,18,19]. As such, our study supports the concept of a secondary, delayed DOX effect on SkM.

Muscle status is influenced by basal activity, food intake and diet composition. In agreement with Diaz-Guerra et al., in administering a cumulative dose of 25 mg/kg DOX over 5 weeks to mice, we observed a significant decrease in basal activity following DOX [25]. However, it is unlikely that the reduction in lean body mass can be fully explained by the decreased basal activity. Despite the halving of activity, it is questionable whether mice truly exhibit sedentary behavior similar to patients with cancer, as they remain relatively active, which may provide some protection against muscle-degrading effects [26]. Moreover, Diaz Guerra et al. demonstrated basal activity normalization over time, despite ongoing cardiotoxicity [25]. Additionally, we observed similar food intake, with a diet relatively high in protein (24 kJ% protein, 9 kJ% fat and 67 kJ% carbohydrates), which may have supported SkM mass preservation [27,28].

Metabolic derangements following DOX were evidenced by IC and RT-qPCR. IC showed decreased EE and a trend toward decreased carbohydrate oxidation following DOX. The RT-qPCR indicated the differential expression of several metabolic markers in WAT and SkM, warranting further validation at the protein or functional level.

Consistent with previous in vitro and in vivo studies, WAT from DOX-treated mice exhibited a steep decrease in *Pparg* expression [15,16]. Conversely, *Ppard* expression remained unaltered following DOX. These two *Ppar* isoforms are well-established transcription factors with distinct tissue distribution and roles in glucose and lipid metabolism. *Pparg,* predominantly expressed in WAT, regulates adipogenesis, lipid storage and whole-body insulin sensitivity, while *Ppard*, broadly expressed across tissues and particularly in SkM, promotes fatty acid (FA) oxidation [29,30,31]. Exploring the PPAR pathway, we found marked decreases in *Cd36* and *Glut4* expression, the primary transporters responsible for FFA and glucose uptake, respectively [32,33]. As such, the DOX-induced WAT loss is likely driven by impairments in adipogenesis and lipogenesis, resulting from reduced FFA and glucose uptake, rather than changes in FA oxidation.

Intriguingly, we observed an opposite response to DOX in SkM. As FFA and glucose uptake in WAT started to decrease, SkM likely attempted to compensate by increasing the expression of *Cd36* and *Glut4* in the GAS. This contrasts with de Lima Junior et al., who reported decreased messenger RNA (mRNA) and protein levels of *Glut4* in the extensor digitorum longus muscle of DOX-treated rats [34]. Conversely, Supriya et al. and Sin et al. found no effect of DOX on *Glut4* protein levels in the GAS of mice [35,36]. It is important to note that these prior studies used a single high-bolus DOX injection (15–18 mg/kg) to examine acute effects, complicating comparison with our findings. However, recently, Osama et al. demonstrated decreased *Glut4* mRNA expression in the GAS of rats undergoing DOX treatment for 2 weeks (15 mg/kg cumulative) [37]. Despite indications of increased fat and glucose uptake, *Ppard* expression remained unchanged following DOX in the GAS. Conversely, carbohydrate oxidation decreased, as suggested by IC and supported by the upregulation of *Pdk4* expression in GAS. *Pdk4* is involved in SkM metabolic fuel switching from carbohydrate to FA oxidation [38,39]. Elevated *Pdk4* levels are associated with metabolic inflexibility and subsequent fat accumulation, marked by the muscle’s impaired ability to switch between fuel sources, often linked to diabetes and obesity [31,38,39]. Accordingly, the ensuing intramyocellular glucose excess is likely channeled into lipid synthesis in addition to being stored as glycogen [30]. Furthermore, experimental studies have shown a direct role of *Pdk4* in promoting the UPS in vitro and in vivo [38,40]. As such, heightened *Pdk4* expression has been linked to conditions associated with muscle atrophy, including animal models of diabetes, cancer cachexia, sepsis, statin-induced myopathy and amyotrophic lateral sclerosis [38,39,40,41]. In line with this, we observed an upregulation of *Murf1* and *Atrogin1* levels in the GAS in response to DOX, two key mediators of the UPS [42].

Hence, it may be likely that following WAT loss, SkM takes over WAT’s trait of being an energy reservoir, in addition to being one of its own, by increasing the uptake of FFA and glucose to enhance their clearance from the bloodstream. However, when an increased uptake rate is not matched by a corresponding rise in oxidation, as suggested by our results following DOX, the resulting imbalance may lead to myosteatosis (i.e., intramuscular fat accumulation), impairing SkM functionality. Eventually, this imbalance could contribute to muscle atrophy. An important consideration in adopting this hypothesis is transporter functionality, which was not assessed but may have been impaired. However, the IC data, along with the other metabolic markers and literature linking cancer therapy in both patients and survivors to myosteatosis, strengthen our hypothesis [2]. Finally, SkM and AT exhibit endocrine functions by secreting myokines and adipokines, respectively [11,43]. Consequently, although beyond the scope of this study, alterations in their expression or secretion profiles could represent another potential contributing mechanism (Figure 6). Besides this, the role of AMP-activated protein kinase (AMPK), a critical energy sensor that promotes energy-generating processes such as glucose and fatty acid uptake and activates catabolic pathways like autophagy and the UPS associated with muscle atrophy, warrants further investigation in this context where DOX has been reported to act as a potent AMPK inhibitor [34,44].

Although DOX increased several markers of protein degradation, a histological evaluation of the CSA revealed a minor tendency toward reduced type IIa fiber size in both the TIA and SOL. However, similarly to Mallinson et al., when a rat model of simvastatin-induced myopathy was used, proteolysis was upregulated before histopathological muscle damage became evident [45]. Moreover, DOX appeared to induce a fast (type II) to slow (type I) fiber type shift in the SOL, although this trend did not reach statistical significance. These results are consistent with findings from other catabolic conditions in mice and humans, including diabetes, sepsis, cancer cachexia and aging [46]. Based on the literature, type II fibers undergo a more pronounced upregulation of the UPS, leading to preferentially type II fiber atrophy and a fast-to-slow fiber type switch. Conversely, muscle disuse primarily causes type I fiber atrophy with a slow-to-fast fiber type switch. Additionally, the same fiber type can respond differently according to muscle type, as previously demonstrated in humans with spinal cord injury [20,46].

This is the first study to examine the combined effects of DEXRA on AT and SkM metabolism. Our findings indicate that DEXRA did not prevent or counteract the DOX-induced reduction in total body mass. In general, DEXRA was ineffective in preventing or reversing any AT-related changes caused by DOX. Regarding SkM, we observed a visual trend toward higher muscle-weight-to-tibia-length ratios in the TIA and SOL of DOX-treated mice following DEXRA administration, approaching control levels. Furthermore, DEXRA seemed to counteract the DOX-induced shift from fast-to-slow fiber types in the SOL by promoting a transition toward a greater proportion of type IIb/x fibers. Concerning the CSA, DEXRA administration in the SOL resulted in a decreased size of type IIb/x muscle fibers in DOX-treated mice, a phenomenon for which we currently lack an explanation. Overall, our findings do not support the initial hypothesis, as DEXRA did not meaningfully prevent or reverse the DOX-induced changes consistently. Importantly, DEXRA was effective in preventing the deterioration of cardiovascular parameters by DOX, as previously reported [23].

### Study Limitations

Although the present study provides valuable insights, we acknowledge some limitations. First, we exclusively used male mice, as they encompass increased sensitivity to DOX-induced CTRCD and are predominantly used in the literature [47,48,49]. Second, the DOX administration route differed from the clinical setting, where administration is performed intravenously. Third, the current study did not include a cancer model, making it difficult to translate the molecular findings to the clinical oncology setting. Fourth, while we evaluated the mRNA expression of several metabolic markers in specific fat and muscle types, other markers and pathways may be involved. Future research should aim to reduce heterogeneity in the selection of muscle types, as seen in our study, and prioritize consistency in the analyzed muscle to deepen discussion and enable more comprehensive conclusions. Aside from the different muscle types, future research should include different fat types (e.g., brown vs. white and subcutaneous vs. epididymal) as they may respond differently. Furthermore, protein levels and functionality should be assessed in addition to gene expression. Finally, the macronutrient composition of food can be a confounding factor and should be carefully considered.

## 4. Material and Methods

### 4.1. Animal Model

A total of 32 10-week-old male C57BL6/J mice (Charles River, France) were housed under standard conditions (20–24 °C, 45–65% relative humidity, 12:12 h light/dark cycle) in the animal housing facility of the University of Antwerp with access to water and regular chow ad libitum. The Ethical Committee for Animal Testing of the University of Antwerp (approval nos. 2021-19) approved all procedures, which conformed to the ARRIVE guidelines and to the EU Directive (2010/63) on the protection of animals used for scientific purposes.

### 4.2. Experimental Design

After one week of acclimatization, mice were randomly divided into four groups (Figure 1A): (1) DOX (*n* = 8), (2) DOX-DEXRA combined (*n* = 8), (3) DEXRA (*n* = 8) and (4) control (CON; *n* = 8). DOX (Adriamycin®, 2 mg/mL, Pfizer, Belgium) was administered via intraperitoneal injection once weekly (4 mg/kg) for 6 weeks (cumulative dose of 24 mg/kg), with DEXRA (Tocris Bio-Techne, Dublin, Ireland)) given 30 min prior to each DOX injection, also via intraperitoneal injection, once weekly (40 mg/kg) for 6 weeks (cumulative dose of 240 mg/kg). CON mice received a vehicle solution (10 mL/kg of a 0.9% NaCl solution; B. Braun, Machelen, Belgium). Following the 5th dose of DEXRA and/or DOX, mice were subjected to indirect calorimetry (IC) measurements in a metabolic cage. After IC measurement, mice received their 6th dose. Finally, 2 days after injection, mice were sacrificed for blood and tissue sampling. Mice treated with DOX were classified as pre-cachectic (≤10% body weight loss) or cachectic (>10% body weight loss) in analogy with previous studies [50,51]. Cardiovascular parameters of the same cohort were reported previously, showing a protective effect of DEXRA [23].

### 4.3. Indirect Calorimetry (IC)

Mice were individually assessed for 4 or 24 h inside the ‘home cage’ of the PhenoMaster Indirect Calorimetry System (TSE, Bad Homburg, Germany). Food intake was recorded by means of a hanging feeder, while physical activity was registered by infrared beam breaks along the X- and Y-axes (X + Y counts). The energy expenditure (EE) and respiratory exchange ratio (RER) were determined through the evaluation of oxygen consumption (VO_2_) and carbon dioxide production (VCO_2_). Additionally, established formulas from the literature were applied to estimate carbohydrate and lipid oxidation [52,53]. Notably, protein oxidation could not be directly assessed from respiratory exchanges, as the oxidation of proteins is incomplete due to urinary nitrogen waste [52,54]. The system utilizes a reference and home cage (single housing), both supplied with ambient air (0.35 L/min flow rate and 0.25 L/min sample flow). Before measurements, the gas analyzer was calibrated against 100% N_2_ and two mixtures of different O_2_ and CO_2_ concentrations.

### 4.4. Blood and Tissue Sampling

Mice were sacrificed by exsanguination after a single intraperitoneal injection of sodium pentobarbital (200 mg/kg, Sanofi, Machelen, Belgium). Blood was collected through the orbital sinus, centrifuged (8000 rpm, for 2 min, at 4 °C) to obtain plasma and stored at −80 °C for molecular analyses. Next, epididymal white adipose tissue (eWAT), Gastrocnemius (GAS), Tibialis anterior (TIA) and Soleus (SOL) muscles from both hindlimbs were harvested and weighed (indexed to tibia length) [55]. The TIA and SOL were used for histological purposes due to their distinct fiber type composition (fast-twitch glycolytic fibers vs. slow-twitch along with fast-twitch oxidative glycolytic fibers, respectively). The GAS, with its mixed fiber type composition, was used for molecular analysis. To this end, the muscle sample of one hindlimb was snap-frozen in liquid nitrogen and stored at −80 °C for molecular analyses, while the other was positioned on a cork, mounted in a Neg-50 Frozen Section Medium (Fisher Scientific, Merelbeke, Belgium) and frozen in liquid nitrogen-cooled isopentane (2-methylbutane) (Honeywell, Charlotte, NC, USA) for immunohistochemical processing. The eWAT was exclusively used for RT-qPCR due to limited tissue availability.

### 4.5. Immunohistochemistry

Myosin Heavy Chain (MyHC) staining was performed to determine the number of total muscle fibers, muscle fiber type-specific composition (type I and IIa) and cross-sectional area (CSA). Paraformaldehyde-fixated muscle (TIA and SOL) cross-sections (10 μm) were air-dried at room temperature, blocked with horse serum and incubated overnight at room temperature with the following primary antibodies: anti-laminin (diluted 1:500, NB300-144, Bio-Techne, Minneapolis, MN, USA), MyHC type I (diluted 1:5, BA-D5, Developmental Studies Hybridoma Bank (DSHB), Iowa City, IA, USA) and MyHC type IIa (diluted 1:5, SC-71, Developmental Studies Hybridoma Bank (DSHB), Iowa City, IA, USA). The day after, slides were incubated for one hour at room temperature with the corresponding immunoglobulin-specific secondary antibodies: Alexa fluor 488 (diluted 1:800, A11008, Invitrogen, Carlsbad, CA, USA) immunoglobulin G (IgG), Alexa fluor 555 (diluted 1:800, A21147, Invitrogen) IgG2b and Alexa fluor 633 (diluted 1:800, A21126, Invitrogen) IgG1. Following incubation, slides were covered with ‘Vectashield ^®^ Antifade Mounting Medium with DAPIT’ (H-1500, Vector laboratories, Burlingame, CA, USA) and air-dried. Digital images were captured at 10× magnification using the NIKON Eclipse Ti series high-throughput fluorescent microscope. Digitally captured images were analyzed blind using ImageJ software v1.54c.

### 4.6. RNA Extraction and Reverse-Transcription Quantitative PCR (RT-qPCR)

Total RNA was extracted from eWAT and the GAS using the RNeasy ^®^ Lipid Tissue Mini Kit (74804, Qiagen GmbH, Hilden, Germany) or the RNeasy ^®^ Midi Kit (75144, Qiagen GmbH, Germany), respectively, according to the manufacturer’s instructions. RNA purity and concentration were evaluated with the NanoDrop ND-1000 spectrophotometer. Relative RNA expression was quantified by performing two-step RT-qPCR using the TaqMan^TM^ Reverse Transcription Reagents (N8080234, ThermoFisher, Waltham, MA, USA), TaqMan^TM^ Universal PCR Master Mix (4304437, ThermoFisher, USA) and Taqman^TM^ primers (see below). *Rpl19* (Mm02601633_g1) and *β-actin* (Mm02619580_g1) were used as consecutive genes for the data normalization of eWAT. *Gapdh* (Mm99999915_g1) and *Rplp0* (Mm00725448_s1) were used as constitutive genes for the data normalization of GAS. Target genes included the following: *Murf1* (Mm01185221_m1), *Atrogin1* (Mm00499523_m1), *Pparg* (Mm00440940_m1), *Ppard* (Mm00803184_m1), *Cd36* (Mm00432403_m1), *Slc2a4/Glut4* (Mm00436615_m1) and *Pdk4* (Mm01166879_m1). RT-qPCR was performed using the QuantStudio^TM^ 3 instrument (ThermoFisher, USA). Relative gene expression was evaluated according to the 2 ^(−∆∆CT)^ method.

### 4.7. Statistical Analysis

Differences between groups were analyzed with a Kruskal–Wallis test followed by Dunn’s multiple comparisons (non-normal distribution) or a two-way ANOVA followed by a Tukey correction (normal distribution) for the general analysis and RT-qPCR data. Normality was assessed through visual inspection (i.e., QQ plot) and a Shapiro–Wilk test. Two-way ANCOVA was used to analyze the IC data with cage time (and AT and SkM) as a covariate, followed by a Bonferroni correction. Differences were considered statistically significant when *p* < 0.05. Data are expressed as the mean ± standard error of the mean (SEM). Statistical analyses were performed in GraphPad Prism v9.3.1. and SPSS v29.0.1.0. Graphs were constructed in GraphPad Prism.

## 5. Conclusions

To the best of our knowledge, this study is the first to examine the concurrent impact of long-term DOX treatment on AT and SkM, while also evaluating the therapeutic potential of DEXRA. Our data demonstrate that DOX induces (pre-)cachexia, characterized by significant AT loss and moderate SkM loss, driven by distinct mechanisms. DEXRA, while effective in preventing DOX-induced CTRCD, failed to prevent the catabolic effects of DOX on AT and SkM. Future studies should explore alternative therapeutic strategies targeting both AT and SkM in cancer cachexia management. 

## Figures and Tables

**Figure 1 ijms-26-01177-f001:**
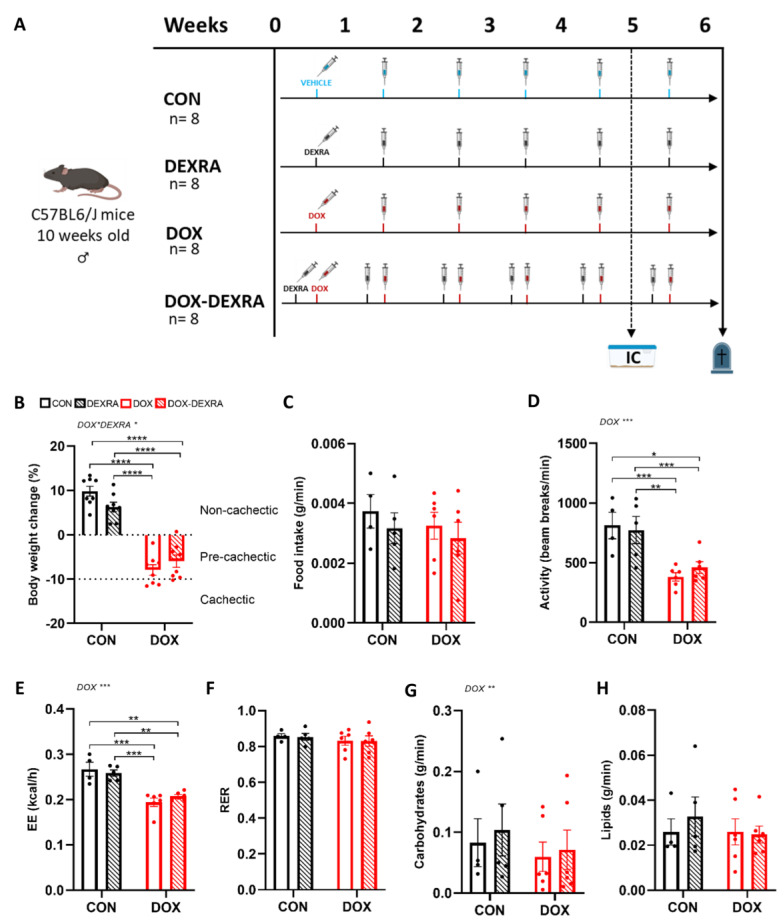
**Experimental design, body weight, food intake, basal activity and metabolic function.** Schematic overview of the study design. Mice were randomly allocated to one of four groups: control (CON, *n* = 8), dexrazoxane (DEXRA, *n* = 8), doxorubicin (DOX, *n* = 8) and doxorubicin combined with dexrazoxane (DOX-DEXRA, *n* = 8). Mice received weekly intraperitoneal injections of DOX (4 mg/kg) and/or DEXRA (40 mg/kg) for a total of 6 weeks. DEXRA was administered 30 min in advance of the DOX injections. Control mice received a vehicle solution (10 mL/kg). In week 5, indirect calorimetry (IC) measurements were performed inside metabolic cages (individual housing). Following IC, mice received their last injection(s). Two days after (week 6), mice were euthanized for blood and tissue collection (**A**). Total body weight change (%) between baseline (week 0) and final measurement (week 6) (**B**). Food intake of mice expressed in grams per minute following the 5th dose, but prior to the 6th dose, of the respective treatment (**C**). Basal activity represented as total beam breaks per minute (**D**). Energy expenditure (without TIA and eWAT weights as covariates) expressed in kilocalories expended per hour (**E**). Respiratory exchange ratio (RER) representing the volume of CO_2_ production over O_2_ consumption (VCO_2_/VO_2_) (**F**). Carbohydrate (**G**) and lipid (**H**) oxidation expressed in grams per minute. Data are represented as bar graphs expressing mean ± SEM. Individual values are additionally shown. Differences between groups were analyzed using two-way ANOVA with the Tukey multiple comparison test (**B**–**D**) or two-way ANCOVA with the Bonferroni multiple comparison test (**E**–**H**). Group sizes: CON—*n* = 4/8; DEXRA—*n* = 5/8; DOX—*n* = 6/8; DOX-DEXRA—*n* = 6/8. * *p* < 0.05, ** *p* ≤ 0.01, *** *p* ≤ 0.001, **** *p* < 0.0001.

**Figure 2 ijms-26-01177-f002:**
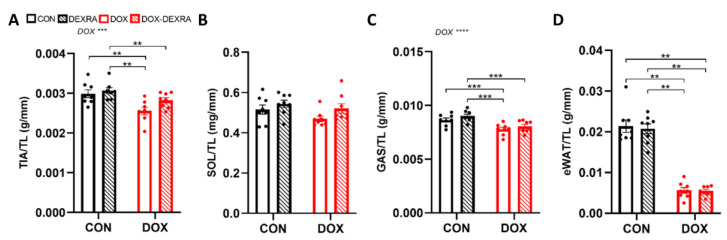
**Skeletal muscle and adipose tissue weight.** Muscle wet weights (mg or g) normalized over tibia length (mm) (TL) for Tibialis anterior (TIA) (**A**), Soleus (SOL) (**B**) and Gastrocnemius (GAS) (**C**) at sacrifice (week 6). Epididymal white adipose tissue (eWAT) wet weight (g) normalized over tibia length (mm) (TL) (**D**) at sacrifice (week 6). Data are represented as bar graphs expressing mean ± SEM. Individual values are additionally shown. Differences between groups were analyzed using two-way ANOVA with the Tukey multiple comparison test (**A**,**C**) or the Kruskal–Wallis test with Dunn’s multiple comparison (**B**,**D**). *n* = 8 per group. ** *p* ≤ 0.01, *** *p* ≤ 0.001.

**Figure 3 ijms-26-01177-f003:**
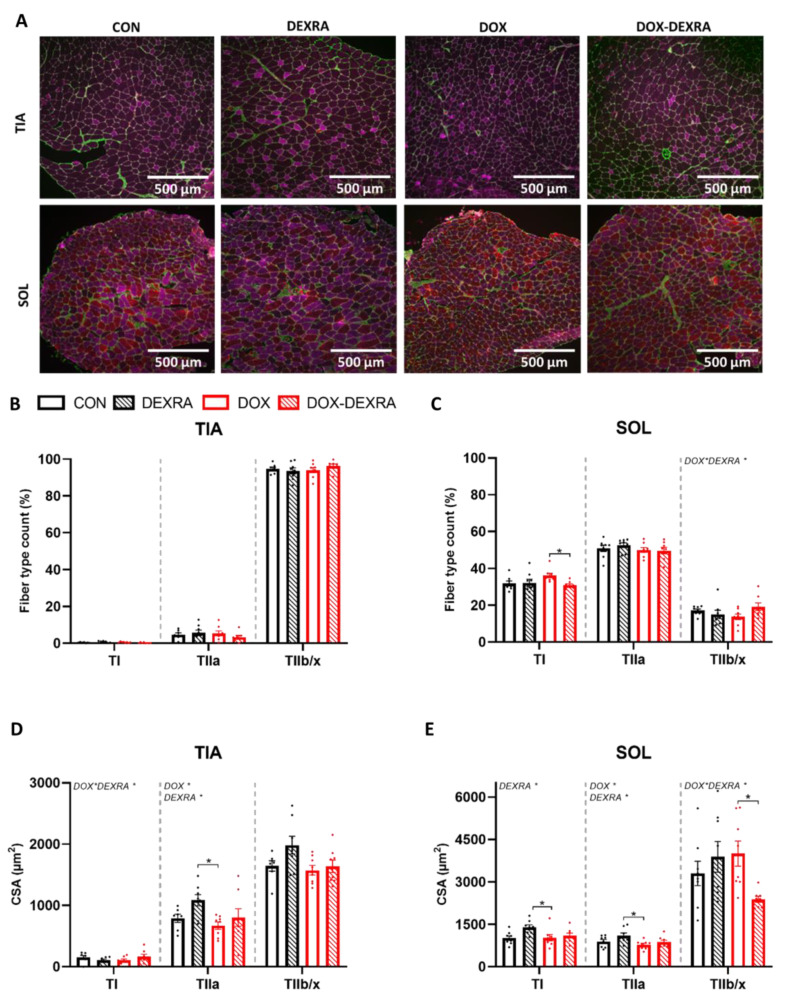
**Muscle fiber type distribution and cross-sectional area (CSA).** Representative images for each group in the immunofluorescence MyHC staining on transversal muscle sections (10 µm) of the Soleus (SOL) and the Tibialis anterior (TIA). Laminin (basal membrane) (green), type I (TI) (red), type IIa (TIIa) (purple) and type IIb/x (TIIb/x) (black—unstained) muscle fibers. Images were obtained at 10× magnification (**A**). Muscle fiber type count of TI, TIIa and TIIb/x fibers, presented as % of total fibers (specific fiber type count/total fiber count) for the TIA (**B**) and SOL (**C**). Muscle fiber type (TI, TIIa and TIIb/x) CSA of the TIA (**D**) and SOL (**E**), presented as the mean individual-fiber CSA (CSA/fiber count). Data are represented as bar graphs expressing mean ± SEM. Individual values are additionally shown. Differences between groups were analyzed using two-way ANOVA with the Tukey multiple comparison test or the Kruskal–Wallis test with Dunn’s multiple comparison. *n* = 7/8 per group. * *p* < 0.05.

**Figure 4 ijms-26-01177-f004:**
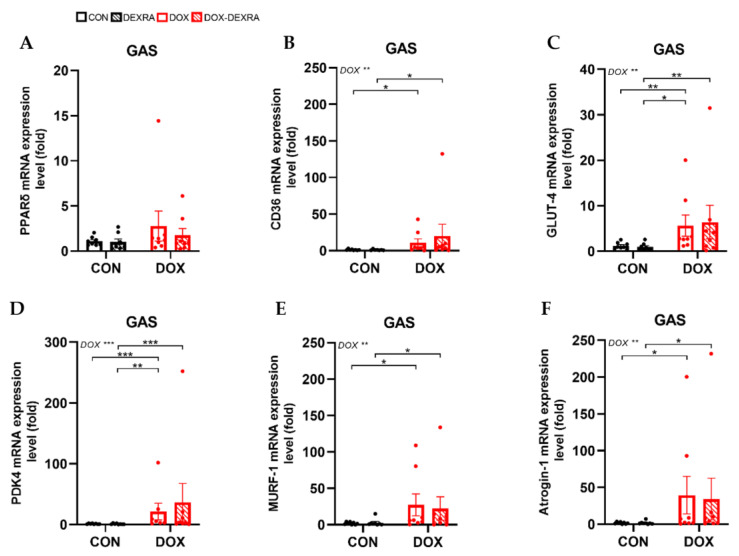
**Molecular changes in skeletal muscle.** Pparδ (**A**), CD36 (**B**), GLUT-4 (**C**), PDK4 (**D**), MURF-1 (**E**) and Atrogin-1 (**F**) gene expression levels in Gastrocnemius (GAS), determined by RT-qPCR. mRNA expression levels are expressed as fold changes over the control group (CON). Results are normalized against GAPDH and RplP0. Data are represented as bar graphs expressing mean ± SEM. Individual values are additionally shown. Differences between groups were analyzed using two-way ANOVA with the Tukey multiple comparison test. Group sizes: CON—*n* = 8; DEXRA—*n* = 7/8; DOX—*n* = 7/8; DOX-DEXRA—*n* = 8. * *p<* 0.05, ** *p* ≤ 0.01, *** *p* ≤ 0.001.

**Figure 5 ijms-26-01177-f005:**
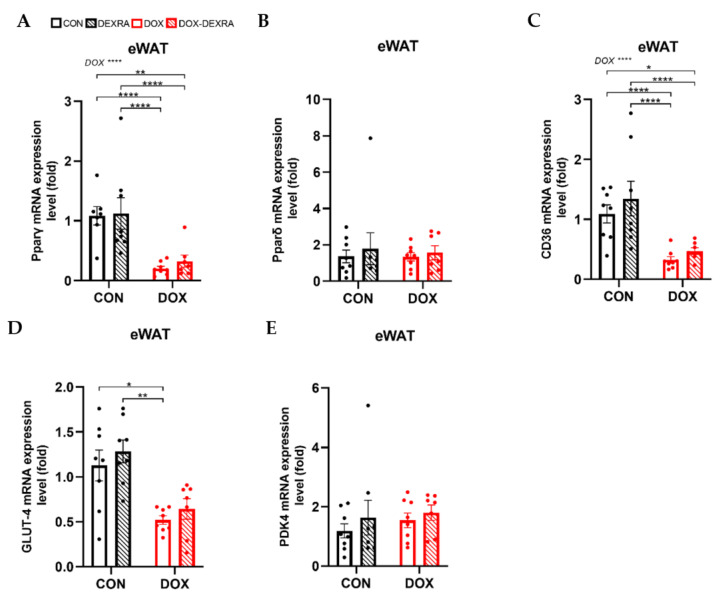
**Molecular changes in adipose tissue.** Pparγ (**A**), Pparδ (**B**), CD36 (**C**), GLUT-4 (**D**) and PDK4 (**E**) gene expression levels in epididymal white adipose tissue (eWAT), determined by RT-qPCR. mRNA expression levels are expressed as fold changes over the control group (CON). Results are normalized against β-actin and Rpl19. Data are represented as bar graphs expressing mean ± SEM. Individual values are additionally shown. Differences between groups were analyzed using two-way ANOVA with the Tukey multiple comparison test (**A**,**C**) or the Kruskal–Wallis test with Dunn’s multiple comparison (**B**,**D**,**E**). Group sizes: CON—*n* = 7/8; DEXRA—*n* = 8; DOX—*n* = 8; DOX-DEXRA—*n* = 7. * *p* < 0.05, ** *p* ≤ 0.01, **** *p* ≤ 0.0001.

**Figure 6 ijms-26-01177-f006:**
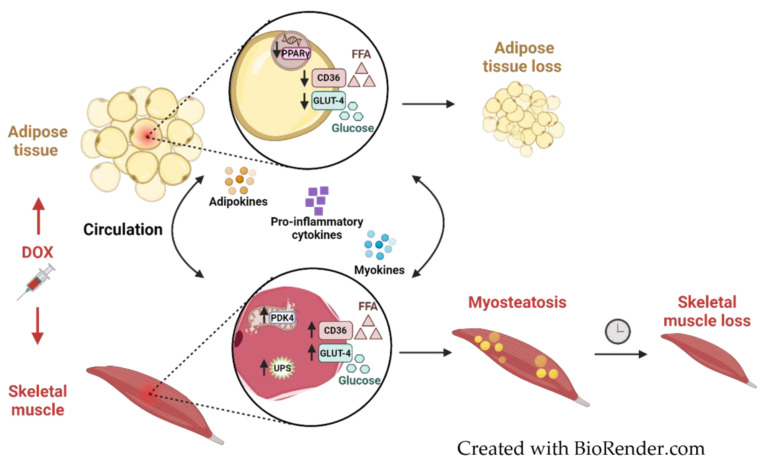
**Conceptual model for DOX-induced adipose tissue and skeletal muscle metabolic impairment and loss.** DOX treatment induces adipose tissue loss by reducing PPARγ expression, which decreases CD36 and GLUT-4 translocation and concomitant FFA and glucose uptake, impairing adipogenesis and lipogenesis. DOX-related skeletal muscle loss results from an imbalance between FFA and glucose uptake and oxidation, as well as impaired fuel oxidation (elevated PDK4 levels), with the subsequent promotion of protein degradation through the ubiquitin–proteasome system (UPS), leading to intramuscular fat accumulation (myosteatosis) and ultimately contributing to muscle atrophy. Myokines and adipokines mediate inter-tissue crosstalk, crucial for regulating tissue homeostasis. Created in BioRender. Van den Bogaert, S. (2025) https://BioRender.com/z40g811 (accessed on 20 January 25).

## Data Availability

The original contributions presented in this study are included in the article. Further inquiries can be directed to the corresponding author(s).

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
