# Peer review of "Distinct Impact of Doxorubicin on Skeletal Muscle and Fat Metabolism in Mice: Without Dexrazoxane Effect"

_ijms, 2025, doi:10.3390/ijms26031177_

Round 1

Reviewer 1 Report

Comments and Suggestions for Authors

In this study, the authors used mice to examine the effects of Dexrazoxane on skeletal muscle atrophy and adipose tissue atrophy caused by doxorubicin. The adverse effects of doxorubicin are major clinical problems. Therefore, clarifying the effect of Dexrazoxane is an important issue.

Major revision

1. The differences in muscle weight are compared in TIA, SOL, and GAS, but the muscle fiber type is compared in TIA and SOL. In addition, the various parameters are verified in GAS, and there is no consistency in the muscles being analyzed. If you have results analyzed using the same muscles, it would be more useful to include them in this figure to deepen the discussion.

2. Regarding eWAT, only mRNA was discussed, and no functional analysis (IHC, WB, etc.) was performed.

3. GLUT4 is a transporter that translocates to the cell membrane and is responsible for the uptake of glucose into the cell. The authors discussed glucose metabolism at the mRNA level, but this is debatable.

Miner comments

1, It would be easier to understand if the experimental design in Figure 1 was inserted at the beginning of Figure 2.

2, The abstract is difficult to read, so please delete the subtitles (background, methods, etc.) and revise it.

3, please insert a scale bar into the image in Figure 4.

Author Response

We wish to thank the reviewer for taking the time to carefully review our manuscript and for providing insightful comments. Point-by-point responses to these comments are provided below.

“In this study, the authors used mice to examine the effects of Dexrazoxane on skeletal muscle atrophy and adipose tissue atrophy caused by doxorubicin. The adverse effects of doxorubicin are major clinical problems. Therefore, clarifying the effect of Dexrazoxane is an important issue.”

Comment 1: The differences in muscle weight are compared in TIA, SOL, and GAS, but the muscle fiber type is compared in TIA and SOL. In addition, the various parameters are verified in GAS, and there is no consistency in the muscles being analyzed. If you have results analyzed using the same muscles, it would be more useful to include them in this figure to deepen the discussion.

We acknowledge that using different muscle tissues for histological and molecular analyses may raise concerns about maintaining a consistent analysis flow. However, the TIA and SOL were selected for histological purposes because of their ease of isolation from origin to insertion and their distinct fiber-type composition. The TIA primarily consists of fast-twitch (type IIb/x—glycolytic) fibers, while the SOL predominantly comprises slow-twitch (type I) fibers along with fast-twitch (IIa—oxidative glycolytic) fibers. These muscles provide the opportunity to study fiber-type-specific responses, which are typically examined through a myosin heavy chain staining. In contrast, the GAS is generally used for molecular analyses due to its larger size and mixed fiber-type composition (fast and slow fibers), allowing for a more comprehensive understanding of the underlying molecular mechanisms. As such, this approach facilitates comparison of results with previous scientific studies that have employed similar methodologies.

We have clarified the rationale for the selection of the muscle tissue in the manuscript in the Material and Methods section (page 13; line 440-443):

“The TIA and SOL were used for histological purposes due to their distinct fiber-type composition (fast-twitch glycolytic fibers vs. slow-twitch along with fast-twitch oxidative glycolytic fibers, respectively). The GAS, with its mixed fiber-type composition, was used for molecular analysis.”

Further, we have addressed the heterogeneity of the studied muscle type by adding it in the “Study Limitations” paragraph in the Discussion section(page 12; line 389-392):

“Future research should aim to reduce heterogeneity in the selection of muscle types, as seen in our study, and prioritize consistency in the analyzed muscle to deepen the discussion and enable more comprehensive conclusions.”

Comment 2: Regarding eWAT, only mRNA was discussed, and no functional analysis (IHC, WB, etc.) was performed.

As correctly raised by the reviewer, functional analyses should additionally be incorporated. However, it is well-established that isolating RNA and protein from mice WAT is challenging. Compared to other tissues, WAT has considerably lower RNA and protein content which is of major concern when dealing with experimental conditions like cachexia where mice exhibit low fat mass.1,2 Similarly, DOX-treated mice from our study experienced significant loss. Given our experience in the lab with the commercial ready-to-use RNA isolation kit from Qiagen specific for lipid tissue, encompassing good RNA purity and concentration, we chose to perform RT-qPCR. However, as already mentioned, due to the limited amount of tissue we used everything for RT-qPCR and unfortunately no tissue was left for functional analyses.

Although this aspect was mentioned as a limitation in the Discussion section (page 12; line 394-395):

“Furthermore, protein levels and functionality should be assessed in addition to gene expression”.

In the revised version, we have added further emphasis to this limitation in (1) the Material and Methods section (page 13; line 447-448) and (2) the Discussion section by inserting a side note to ensure it is explicitly highlighted for readers (page 10; line 289-290):

  • “The eWAT was exclusively used for RT-qPCR due to limited tissue availability”
  • “The RT-qPCR indicated differential expression of several metabolic markers in WAT and SkM, warranting further validation at protein or functional level.”.

Comment 3: GLUT4 is a transporter that translocates to the cell membrane and is responsible for the uptake of glucose into the cell. The authors discussed glucose metabolism at the mRNA level, but this is debatable.

We acknowledge and concur with this comment. However, in the Discussion section we comprehensively address the indirect calorimetry, histological and molecular findings, situating these results within the context of existing literature. Consequently, energy metabolism (carbohydrate —including glucose— and lipid) was discussed based on mRNA levels as well as indirect calorimetry findings. The latter being regarded as the gold standard for measuring energy metabolism in clinical practice, particularly for estimating energy expenditure and substrate (carbohydrates and fat) oxidation.3 However, we acknowledge the reviewer’s concern regarding the need for careful and nuanced interpretation when discussing alterations in glucose and lipid metabolism, particularly in light of the absence of functional analyses.

To provide additional context and nuance to the findings, we have now included the following in the Discussion section (page 10; line 289-290):

”The RT-qPCR indicated differential expression of several metabolic markers in WAT and SkM, warranting further validation at protein or functional level.”.

Comment 4: It would be easier to understand if the experimental design in Figure 1 was inserted at the beginning of Figure 2.

This is an excellent suggestion. We have merged both figures into a single figure (i.e. Figure 1 in the revised manuscript, page 3).

Comment 5: The abstract is difficult to read, so please delete the subtitles (background, methods, etc.) and revise it.

To enhance readability and ensure greater conciseness, we revised the abstract by removing subtitles and descriptive statistics (page 1; line 14-42).

Comment 6: Please insert a scale bar into the image in Figure 4.

We apologize for the oversight in our initial submission. Scale bars have been added (i.e. Figure 3 in the revised manuscript, page 6).

References:

  1. An, Y. A., & Scherer, P. E., “Mouse adipose tissue protein extraction.,” Bio-protocol, vol. 10, no. 11, pp. e3631, June 2020, doi: 10.21769/BioProtoc.3631.
  2. Tan, P., Pepin, É., & Lavoie, J. L., “Mouse adipose tissue collection and processing for RNA analysis.,”, JoVE, no. 131, Jan 2018, doi: 10.3791/57026.
  3. Delsoglio, M., Achamrah, N., Berger, M. M., & Pichard, C., “Indirect calorimetry in clinical practice.,”, J of clin med, vol. 8, no. 9, September 2019, doi: 10.3390/jcm8091387.

Reviewer 2 Report

Comments and Suggestions for Authors

Article provides an insight on mechanisms leading to doxorubicin(DOX)-induced muscular atrophy. The "Introduction" section provides a sound background about doxorubicin's influence on muscular tissue and possible positive effects of dexrazoxane on DOX-induced changes in the muscules. However, article lacks a clear "aim of the study", which should be given (stated) at the end of introcution section. Statement given on p3, L100-105 cannot substitute the aim of the study. 

The "Materials and methods" section provides sufficient details about models and methods used in the study. 

The "Results" section describes the obtained results in sufficient detail and has enough figures for visualisation of the study results.

The "Discussion" section is devoted to explanation of meaning and possible mechanisms of development of observed changes during DOX-induced muscle atrophy and its correction by dexrazoxane. In my opinion, authors should also discuss a possible role of AMP activated protein kinase (AMPK) cascade in observed changes in muscles and fat tissue since doxorubicin is a potent AMPK inhibitor and AMPK plays a crucial role in glucose, free fatty acids and energy metabolism. Also, a limitations of the study should be mentioned at the end of discussion section. 

Conclusions should be reworked according to the "renewed" aim of the study and should be clearer. 

Author Response

We wish to thank the reviewer for taking the time to carefully review our manuscript and for providing insightful comments. Point-by-point responses to these comments are provided below.

“Article provides an insight on mechanisms leading to doxorubicin(DOX)-induced muscular atrophy. The "Introduction" section provides a sound background about doxorubicin's influence on muscular tissue and possible positive effects of dexrazoxane on DOX-induced changes in the muscles.”

Comment 1: The article lacks a clear "aim of the study", which should be given (stated) at the end of the introduction section. Statement given on p3, L100-105 cannot substitute the aim of the study.

We acknowledge that the aim of the study was not clearly stated at the end of the Introduction section. As a result, this section has been rewritten (page 2-3; line 97-100):

The present study aimed to investigate the catabolic effects of DOX on SkM and AT mass and function using a 6-week treatment protocol in mice, as well as to evaluate the therapeutic effects of DEXRA. This study builds on our previous findings that DEXRA protects against DOX-induced vascular toxicity in these mice [23].”

Comment 2: The "Discussion" section is devoted to explanation of meaning and possible mechanisms of development of observed changes during DOX-induced muscle atrophy and its correction by dexrazoxane. In my opinion, authors should also discuss a possible role of AMP activated protein kinase (AMPK) cascade in observed changes in muscles and fat tissue since doxorubicin is a potent AMPK inhibitor and AMPK plays a crucial role in glucose, free fatty acids and energy metabolism. Also, a limitations of the study should be mentioned at the end of discussion section.

With regards to the final comment, a study limitations section was present. To improve visibility and transparency for the reader a “Study Limitations” subtitle was constructed (page 12; line 381-396).

We appreciate the insightful comment regarding the role of AMPK. In fact, we initially overlooked AMPK due to pilot experiments that did not show significant changes in AMPK following DOX on western blot, and because RT-qPCR would not have yielded meaningful insights (phosphorylation of AMPK). However, the reviewer is right that AMPK should be discussed in the context of DOX-induced (pre-)cachexia.1,2 Therefore, we elaborated on the potential role of AMPK in the Discussion section (page 11; line 338-342):

“Besides that, the role of the AMP-activated protein kinase (AMPK), a critical energy sensor that promotes energy-generating processes such as glucose and fatty acid uptake, and activates catabolic pathways like autophagy and the UPS related to muscle atrophy, warrants further investigation in this context where DOX has been reported to act as a potent AMPK inhibitor.

Comment 3: Conclusions should be reworked according to the "renewed" aim of the study and should be clearer. 

The conclusion has been revised with a clear take-home message (key findings) according to the study aim (page 14, line 492-499):

“To the best of our knowledge, this study is the first to examine the concurrent impact of long-term DOX treatment on AT and SkM, while also evaluating the therapeutic potential of DEXRA. Our data demonstrates that DOX induces (pre-)cachexia, characterized by significant AT loss and moderate SkM loss, driven by distinct mechanisms. DEXRA, while effective in preventing DOX-induced CTRCD, failed to prevent the catabolic effects of DOX on AT and SkM. Future studies should explore alternative therapeutic strategies targeting both AT and SkM in cancer cachexia management.” 

References:

  1. Steinberg, G. R., & Hardie, D. G., “New insights into activation and function of the AMPK.,” Nature reviews Mol cell biol, vol. 24, no. 4, pp. 255-272, Apr. 2023, doi: 10.1038/s41580-022-00547-x.
  2. A. de Lima Junior et al., “Doxorubicin caused severe hyperglycaemia and insulin resistance, mediated by inhibition in AMPk signalling in skeletal muscle.,” J Cachexia Sarcopenia Muscle, vol. 7, no. 5, pp. 615–625, Dec. 2016, doi: 10.1002/jcsm.12104.

Round 2

Reviewer 1 Report

Comments and Suggestions for Authors

The authors have appropriately addressed all the items noted.